# Evaluating Self-Pollination Methods: Their Impact on Nut Set and Nutlet Abscission in Macadamia

**DOI:** 10.3390/plants13243456

**Published:** 2024-12-10

**Authors:** Palakdeep Kaur, Max Cowan, Joanne De Faveri, Mobashwer Alam, Bruce Topp

**Affiliations:** 1Queensland Alliance for Agriculture and Food Innovation, The University of Queensland, St. Lucia, QLD 4072, Australia; m.alam@uq.edu.au; 2Department of Primary Industries, Maroochy Research Facility, Nambour, QLD 4560, Australia; max.cowan@daf.qld.gov.au; 3AV Data Analytics, Adelaide, SA 5153, Australia; jdefaveri.avda@gmail.com

**Keywords:** macadamia, autogamy, geitonogamy, self-pollination, self-fertilization, open pollination, initial nut set, final nut set, nutlet abscission

## Abstract

Nut set is an important determinant of yield and plays a pivotal role in orchard profitability. The complex process of nut setting is governed by numerous factors, with pollination being a critical mechanism. Macadamia cultivars exhibit both self- and cross-pollination. Self-pollination may increase nut set, so it is a trait of interest in breeding. This study investigated nut setting and nutlet abscission on four cultivars, ‘HAES 791’, ‘HAES 741’, ‘HAES 344’, and ‘A16’, using three controlled self-pollination methods: (i) autogamy (AG), entailing bagging before anthesis with no hand-pollination; (ii) geitonogamy 1 (GG1), bagging following hand-pollination using pollen from the same raceme; and (iii) geitonogamy 2 (GG2), bagging following hand-pollination using pollen from different racemes of the same cultivar. These self-pollination methods were compared against open-pollination (OP). ‘HAES 741’ and ‘HAES 791’ were partially self-fertile, while ‘HAES 344’ and ‘A16’ were self-infertile. Final nut sets per raceme for ‘HAES 741’ were 0.43 by AG, 0.65 by GG1, and 0.5 by GG2, and for ‘HAES 791’, they were 0.90 by AG, 1.25 by GG1, and 1.0 by GG2. Final nut set per raceme with OP was higher compared to self-pollination methods and ranged between 3.5–6.5. In self-fertile cultivars, nut set following the three self-pollination methods accounted for 6.5–3.7% of the nut set following OP, and nutlet abscission following self-pollination methods accounted for 20–50% of nutlet abscission following OP. No significant differences in nut set and nutlet abscission were observed among AG, GG1, and GG2. Results suggest that macadamia orchards planted with self-fertile cultivars would be less reliant on external pollinators or artificial pollination to achieve adequate yields.

## 1. Introduction

Macadamia belongs to the family Proteaceae and is native to the rainforests of Australia. *Macadamia integrifolia* and *M*. *tetraphylla* are grown commercially for their edible and nutritious kernels. Macadamia is a partially self-incompatible tree crop where most cultivars are self-incompatible; some have the ability to set nuts following self-pollination. The crop depends on cross-pollination for optimal reproductive success [1,2].

Nut formation depends on successful pollination and fertilization, which can be influenced by several factors, including type of pollination and source of pollen [3]. Pollination can be self or cross. Self-pollination is divided into two types: ‘autogamy’(AG), where pollination occurs within the same flower, or ‘geitonogamy’ (GG), in which pollination occurs between different flowers of the same plant [4]. Eckert [4] stated that geitonogamy can occur due to pollination between flowers of the same inflorescence (GG1), between different inflorescences of the same plant (GG2), or between different inflorescences of the same genotype. Like cross-pollination, geitonogamy is dependent upon pollinators. Macadamia produces perfect or hermaphrodite flowers that consist of male (stamen) and female (pistil) reproductive parts on the same flower [5]. The flowers exhibit protandry—the anthers dehisce before the stigma is receptive, so pollen is deposited on the stigma within the unopened flowers [6].

Macadamia has an extremely low fruit to flower ratio [7]. Under optimal environmental and management conditions, a fully grown macadamia tree can produce 2500 racemes (inflorescence) in a blooming season [6]. Each raceme constitutes nearly 300 hermaphrodite florets, and 6–35% of these florets progress into immature nuts. Nevertheless, at maturity, just 0.3% of florets eventually develop into harvestable nuts [7,8]. This abscission of flowers and fruits occurring from anthesis to maturity can be divided into three periods. The first abscission period happens within two weeks of flower opening and mainly comprises pollinated but unfertilized flowers. The second abscission period, termed ‘initial set’ by [5] occurs 3–8 weeks after anthesis and involves young nuts (nutlets). Almost all fruits at this stage are fertilized and contain normal developing embryos. Urata [5] noted that limiting nutritional resources could be a reason behind this premature abscission. According to Sakai and Nagao [8], an increase in ethylene levels leads to fruit abscission at this stage. The third period of abscission occurs 10–30 weeks post-anthesis as nuts accumulate dry weight and total oils. A limitation on photosynthates [9] or increased temperatures during the latter developmental stages could be the reason for this abscission [10].

A 6 × 6 diallel experiment observed that out of 30 cross-combinations, 29 showed significantly higher initial and final fruit sets than self-pollinated racemes [11]. Racemes subjected to supplemental hand cross-pollination set 21.54% more nuts than open-pollinated racemes. The nut abscission rate following an autogamous self-pollination method was high, ranging between 88–100%, compared to 33–75% in open- and cross-pollinated racemes [1]. Supplementary cross-pollination improved the final nut set by 57–97% in ‘246’ [12]. In previous studies, nut set and nut abscission were compared for open-, cross-, and self-pollinations. However, there is a lack of studies elucidating differences in the rates of nut set and nutlet abscission within self-pollination methods.

The ability of a plant to produce seeds without pollinators or pollinizers is proposed as the main advantage of self-pollination [13]. Despite the capacity of some macadamia cultivars to set nuts through self-pollination, there is limited research on whether this pollination would potentially benefit from a pollen vector or if it occurs autonomously. This study examines the effects of three self-pollination methods on the initial and final nut sets and nutlet abscission in four macadamia cultivars.

## 2. Results

### 2.1. Effects of Different Pollination Methods on Nut Set

There was variation in the initial and final nut sets among the four cultivars (Figure 1 and Figure 2). There was a significant difference between the open- and self-pollination methods for ‘HAES 741’ and ‘HAES 791’ at INS and FNS (*p* < 0.05) (Figure 1 and Figure 2). ‘HAES 344’ and ‘A16’ did not set any nuts with self-pollination methods, although the nut set was counted on racemes subjected to open-pollination. In the four cultivars, nut set was the highest on open-pollinated racemes (Figure 1 and Figure 2), with INS and FNS per raceme ranging from 6.5–8.0 ± 0.72 and 3.5–6.5 ± 0.38, respectively. FNSs per raceme following different self-pollination methods were 0.43 ± 0.38 by AG, 0.65 ± 0.38 by GG1, and 0.55 ± 0.38 by GG2 for ‘HAES 741’. The counts were 0.90 ± 0.38, 1.25 ± 0.38, and 1.0 ± 0.38, respectively, for ‘HAES 791’.

No significant difference was observed in the interaction between the self-fertile cultivars and self-pollination method, suggesting that the effects of different self-pollination methods on the nut set are independent of the cultivar (Table 1). There was a significant difference between self-pollination methods (AG, GG1, and GG2) at INS (*p* = 0.019), while no difference was observed at FNS (*p* = 0.273) (Table 1).

Further analysis showed that INS following GG1 and GG2 was significantly higher, compared to AG (Figure 3). A significant difference between methods at INS was observed between AG-GG1 and AG-GG2 but not at FNS (Figure 3). ‘HAES 741’ and ‘HAES 791’ set the highest nuts per raceme in GG1, with 1.62 ± 0.13 at INS and 0.95 ± 0.12 at FNS, followed by GG2 with 1.35 ± 0.13 at INS and 0.77 ± 0.12 at FNS, and AG with 0.87 ± 0.13 at INS and 0.65 ± 0.12 at FNS (Figure 3).

Self-fertile cultivars (‘HAES 741’ and ‘HAES 791’) differed significantly at INS (*p* < 0.001) and FNS (*p* = 0.009) (Table 1). ‘HAES 791’ set a significantly higher number of nuts with all self-pollination methods, compared to ‘HAES 741’ (Figure 1 and Figure 2). ‘HAES 791’ set a significantly higher number of nuts following autogamy at INS and FNS, suggesting that ‘HAES 791’ is more self-fertile than ‘HAES 741’ (Figure 4).

### 2.2. Effects of Different Pollination Methods on Nutlet Abscission

A significant difference between self-pollination methods at INS (*p* = 0.019) and between self-fertile cultivars at both INS (*p* < 0.001) and FNS (*p* = 0.009) (Table 2) prompted a subsequent analysis of nutlet abscission. Self-pollination methods (AG, GG1, and GG2) were not significantly different for nutlet abscission (*p* = 0.222) (Table 2). No differences were observed between self-fertile cultivars (‘HAES 741’ and ‘HAES 791’) (*p* = 0.062) and for cultivar and method interaction (*p* = 0.482).

Self- and open-pollination methods were significantly different for nutlet abscission (*p* = 0.035) (Table A1). Nutlet abscission per raceme was highest following OP (Figure 5). It ranged between 1.45–3.95 for the four cultivars and was highest for ‘HAES 791’ (3.95 ± 0.59). In ‘HAES 344’ and ‘A16’, as there was no nut set following self-pollination methods, nutlet abscission was observed following OP (Figure 5). Non-significant nutlet abscission following GG1 and GG2 was higher than AG (Figure 5). For ‘HAES 741’, mean nutlet abscissions per raceme were 0.25 ± 0.79 for AG, 0.36 ± 0.79 for GG1, and 0.83 ± 0.79 for GG2. For ‘HAES 791’, they were 0.82 ± 0.71, 1.72 ± 0.71, and 1.24 ± 0.71, respectively.

## 3. Discussion

Due to partial self-incompatibility in macadamia, the nut set varies among cultivars following self-pollination. Usually, productivity in orchards is improved by the inclusion of pollinators [1,14,15].This study evaluated the effects of three self-pollination methods, including pollinator-independent autogamy (AG) and pollinator-dependent geitonogamy (GG1 and GG2), on nut set and nutlet abscission. It was hypothesized that geitonogamy (pollinator-dependent) would result in a higher nut set than autogamy (pollinator-independent). Firstly, there is a spatial separation of anthers and stigmas within florets, preventing autogamy; secondly, florets exhibit protandry, with different maturity periods between reproductive parts within florets, increasing the chance of geitonogamy; and thirdly, a mature tree produces approximately 2500 racemes each, with 200–250 florets [6] and multiple florets open each day. As a result, geitonogamy can occur within the same raceme, between different racemes of a tree, or between racemes of different trees of the same cultivar. 

The results confirmed the partial self-fertility of cultivars ‘HAES 741’ and ‘HAES 791’, which produced 0.55 ± 0.42 and 1.0 ± 0.42 self-pollinated nuts per raceme, respectively, as observed in previous research [1,16]. ‘HAES 344’ and ‘A16’ produced no self-seeds, classifying them as self-infertile. There was no difference in nut set and nutlet abscission between the three self-pollination methods (AG, GG1, and GG2) (Figure 1, Figure 2 and Figure 3). However, there was a significant difference between self- and open-pollination for nut set and nutlet abscission (Figure 1, Figure 2 and Figure 3). Among the different pollination methods used, nut set was highest with open-pollination. Similar results have been reported in macadamia by [1,15] and in other crops, such as plum (*Prunus salicina* Lindl.) [17] and mango (*Mangifera indica*) [18]. This could be due to the availability of genetically diverse pollen, which increases the likelihood of successful fertilization, leading to a higher nut set [19]. Conversely, self-pollination methods are limited by the genetic uniformity of pollen, which can reduce fertilization efficiency because of the partial self-incompatibility mechanism in macadamia. Nut set following the three self-pollination methods constituted 6.5–10.5% of the nut set following OP for ‘HAES 741’ and 25–34.7% for ‘HAES 791’ (Figure 2). In self-fertile cultivars, nutlet abscission following autogamy accounted for nearly 20% of nutlet abscission following OP, while in self-infertile cultivars, abscission was recorded from the racemes tagged for OP only (Figure 5). Similarly, Howlett, Read [15] also reported that hand self-pollination did not increase the final nut set in macadamia, but they did not report on nutlet abscission. In GG2, where pollen was transferred from a different raceme of the same tree, a higher nutlet abscission rate is expected—pollen viability may be affected during transportation from one raceme to another, whereas in GG1, pollen transfer occurs within the same raceme. In addition, macadamia exhibits protandry, and stigmas are receptive 2–3 days after anthesis [6]. Therefore, there is a possibility that some stigmas may not be receptive when viable pollen from one raceme is transferred to another raceme in GG2, leading to unsuccessful pollination and fertilization and, ultimately, higher abscission. In GG1, the chances of florets with different stigma receptivity times at pollination are lower.

Unlike other crops, where either autonomous self-pollination could not set fruit [13,20] or pollinator/hand-mediated self-pollination resulted in a significantly higher fruit set [21,22,23], this study focused on cultivars from two fertility groups and showed that in self-fertile cultivars, self-pollination can occur, even in the absence of pollinators or artificial pollination. Conversely, self-infertile cultivars cannot set self-nuts, whether with pollen from the same raceme or from a different raceme of the same tree. Based on the results of this study, the following suggestions can be made: First, cross-pollination increases yield, as OP yielded higher nut sets in all four cultivars. Second, in macadamia orchards established solely with self-fertile cultivars, pollinator-dependent (geitonogamy) and independent self-pollination methods (autogamy) may produce similar yields because there was no significant difference between AG, GG1, and GG2 at final nut set. Third, the autogamous self-pollination method is sufficient to identify the self-fertility of cultivars in future studies. This is particularly important in breeding because screening for self-fertility using autogamy is less expensive than geitonogamy. Geitonogamy requires the re-opening of bags and the application of the self-pollen, whereas screening for autogamy does not. Fourth, as each macadamia floret consists of male and female reproductive parts, autogamy eliminates the need to allocate resources to find potential pollen donors. Fifth, abscised nuts comprise a mixture of self- and cross-pollinated nuts because in both self-fertile and self-infertile cultivars, nutlet abscission was observed following OP.

Self-fertile cultivars can reduce the dependency on external pollinators and pollinizers, thereby reducing labor and management costs associated with the installation and maintenance of beehives [24]. Furthermore, self-fertility provides the possibility of planting a single cultivar in blocks. The synchronization of flowering time and nut maturity within a single block mitigates the need for multiple harvests, simplifying orchard management. Monoculture blocks provide uniform produce, subsequently negating the need for fruit or nut grading processes [25]. On the other hand, in a multi-varietal orchard, the interplanting of cross-compatible cultivars and the timely visitation of pollinators are pre-requisites for fruit set. Additionally, the flowering periods of cultivars must coincide. Even under these optimal conditions, the assurance of fruit or nut set at harvest remains uncertain. Adverse climatic conditions, such as drought, floods, forest fires, and disease infestation, can cause significant losses of flowers and impede pollinator activity [26]. Hence, the incorporation of self-fertile cultivars while establishing a macadamia orchard may be economically beneficial. Partially self-fertile cultivars, such as ‘HAES 741’ and ‘HAES 791’, can be used as parents in breeding programs to incorporate the self-fertility trait into new cultivars, particularly for regions with suboptimal cross-pollination conditions. For the incorporation of the self-fertility trait in new cultivars, screening of germplasm is essential to identify high-yielding, self-fertile cultivars. Phenotypic screening of self-fertility is time-consuming and labor-intensive. The identification of molecular markers will allow breeders to select cultivars with the desired trait at an early stage, thus streamlining the breeding process [27]. By demonstrating that self-fertile cultivars may lead to higher and consistent yields, future research could provide a compelling case for growers to adopt these cultivars, thereby, optimizing orchard productivity. This experiment has limitations. Firstly, although racemes were bagged to avoid external pollinators in self-pollination treatment groups, pollen transfer could still occur within bags, for example, facilitated by air movement or the activity of small insects. Consequently, it is challenging to differentiate between intra- and inter-floret pollination within the same raceme. To minimize the contamination of the particular pollination method, fine mesh bags are recommended over paper bags, due to their superior durability under various weather conditions. Paper bags are susceptible to damage from rain and can easily tear. During the experiment, bags were regularly checked to ensure they remained intact. This regular monitoring was crucial to maintain the reliability of the pollination process. Secondly, the raceme selection process involved the random selection of reachable racemes and those at the required developmental stage (looping). Further study using stratified sampling of all regions of the tree would be required to determine if bias was introduced using the current methodology. Additionally, experimenting on the entire tree structure will provide more accurate and holistic insights. Thirdly, there were two cultivars (‘HAES 741’ and ‘HAES 791’) that were identified as self-fertile [1,16]. The current study is part of an ongoing research work on self-fertility in macadamia. A broader range of genotypes identified as self-fertile should be included in future studies to provide further insight into the control of self-pollination in macadamia.

Macadamia florets are partially self-incompatible and primarily pollinated by honeybees and stingless bees [28]. Primary production of cross-pollinated crops is limited by a reduction in the number and diversity of bees globally [29,30,31]. Accelerated use of pesticides, chemicals, land-use transformation, and climate change have lethal effects on the lives of bees. Given macadamia’s dependency on pollinators [2,15], a decline in pollinator populations may have serious implications for future yields. Consequently, the incorporation of self-fertility has become one of the objectives in macadamia breeding programs [1]. The identification of the mechanisms regulating self-fertility forms an important research goal in macadamia breeding. Macadamia florets are hermaphroditic and protandrous; hence, spatial and temporal separations of the reproductive organs could contribute to the partial self-fertility observed in some cultivars. Furthermore, self-incompatibility is a genetically controlled mechanism that prevents a pollen grain from setting fruit with a maternal plant of the same genotype. Identification of the genes regulating self-fertility will provide insights into the molecular mechanism underlying this trait. The development of self-fertile cultivars has also become a breeding objective in other partially self-compatible crops, such as almond [24,32,33], blueberries [34], apple [35], and mango [36]. While self-fertility is an important trait in breeding, it can complicate hybrid seed production. The current method of producing bi-parental macadamia families does not require emasculation, as seed parents are self-infertile [37]. Using self-fertile seed parents will require emasculation to ensure pure bi-parental families. Macadamia florets are very small, and each raceme contains nearly 300 florets [6], making manual emasculation extremely time consuming. An alternative would be the introduction of an efficient emasculation method or conducting a paternity analysis of progeny developed from self-fertile parents, although paternity analysis can be costly. Another possibility is to cross a self-infertile female with a self-fertile male parent, which can eliminate the need for emasculation. However, selecting for self-fertility in only one parent will reduce the genetic gain by half. This research establishes a foundation for future studies to efficiently incorporate the self-fertility trait in breeding programs.

## 4. Materials and Methods

### 4.1. Study Site and Germplasm

Experimental trees were located in the macadamia arboretum at Maroochy Research Facility, Queensland Department of Agriculture and Fisheries, Nambour QLD (Table 3). All selected trees were approximately 30 years old.

Weather data for the study period were obtained from WillyWeather—Australian Weather Forecast [38]. Data were provided for 5 months from September 2022–January 2023, i.e., the period from pollination to the final nut set count data collection (Table 4). The parameters recorded included monthly maximum temperature (°C) and minimum temperature (°C), total rainfall (mm), and relative humidity (%). The weather station was Nambour (Maroochy Research Facility), where the cultivars were located.

### 4.2. Experimental Design

For each pollination method, 10 racemes per tree were randomly selected at full bloom (September 2022). The methods included (1) autogamy (AG); (2) geitonogamy 1 (GG1); (3) geitonogamy 2 (GG2); and (4) open-pollination (OP). The methods included: (1) bagging of racemes before anthesis to exclude pollinators, autogamy (AG); (2) bagging of racemes following hand pollination with pollen from the same raceme, geitonogamy 1 (GG1); (3) bagging of racemes following the removal of self-pollen and hand pollination with pollen from a different raceme of the same tree, geitonogamy 2 (GG2); and (4) open-pollination (OP)—racemes were tagged but not bagged and allowed to undergo natural crossing by open pollen sources (control). In autogamy, the raceme is considered the unit of flower rather than individual florets, as it is likely that selfing occurs within a floret rather than between different florets in the absence of pollinators. Racemes were enclosed in 30 cm × 15 cm brown paper bags at the ‘looping’ stage of the floret; looping is defined as being 2–3 days before anthesis, where the style forms a loop in its middle section [12,39] to prevent contamination by outcross pollen. To hand-pollinate racemes using the methods GG1 and GG2, pollen was collected from the donor raceme with a clean plastic tube (2–2.5 cm in diameter and 25 cm long) by rubbing the inner surface against previously bagged and freshly opened racemes [1]. Pollen grains were visible on the inner surface of the tube with the naked eye. The racemes to be pollinated were positioned in the tube, and the tube was rotated, ensuring pollen contact with the stigma.

Initial nut set (INS) was evaluated at 4 weeks post-anthesis. Final nut set (FNS) counts were conducted 14–16 weeks post-anthesis. Every experimental raceme was tagged with a specific bag number, pollination method, and initial and final nut set counts (Figure 6).

### 4.3. Statistical Analysis

Data collected from each tree were analyzed using REML (Restricted Maximum Likelihood) Linear Mixed Model in GenStat-21 [40]. To understand the effects of different pollination methods on nut set and nutlet abscission, a Wald test was used to assess the overall treatment effect. Fisher’s protected least significant difference test was used for multiple comparisons. The model included fixed-effect terms for the cultivar and methods and their interaction and the random-effect terms for tree identity number to capture the fact that different pollination methods were nested within each tree. For nutlet abscission, the difference between INS and FNS per raceme was calculated.

## 5. Conclusions

Nut set in macadamia orchards is highly dependent on cross-pollination. This study investigated the effects of self-pollination methods on two self-fertile and two self-infertile cultivars. The currently tested self-fertile cultivars did not achieve adequate yields solely through self-pollination. Cultivars exhibiting self-fertility traits can offer a number of advantages over self-infertile varieties, such as stable yield under low or absent external pollinator/pollinizer populations and under asynchronous flowering conditions. These advantages warrant further research on the self-fertility trait in macadamia breeding. A study evaluating the effects on a wider variety of cultivars is required. Investigating the mechanism regulating this trait, as well as identifying the molecular markers associated with the trait, is a further goal that can help in understanding the self-fertility trait in macadamia and other crops.

## Figures and Tables

**Figure 1 plants-13-03456-f001:**
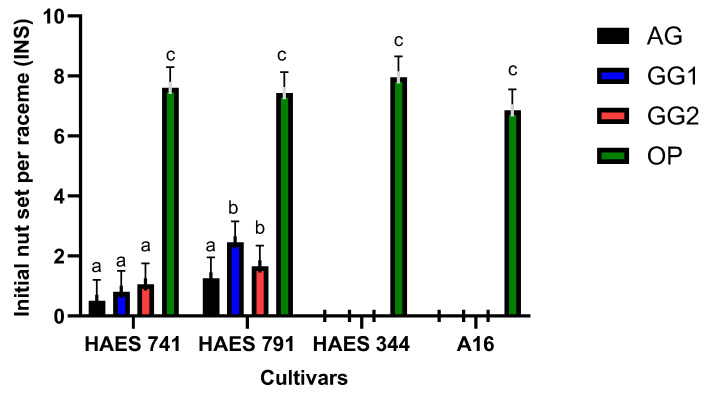
Initial nut set (INS) per raceme of all cultivars following different pollination methods: autogamy (AG), no hand-pollination, bagged; geitonogamy 1 (GG1), hand-pollination within the raceme, bagged; geitonogamy 2 (GG2), hand-pollination within the cultivar, bagged; open-pollination (OP), control. Bars indicate the mean INS per raceme (±1 standard error) for the respective pollination methods. Bars marked with different letters are statistically significant (*p* < 0.05).

**Figure 2 plants-13-03456-f002:**
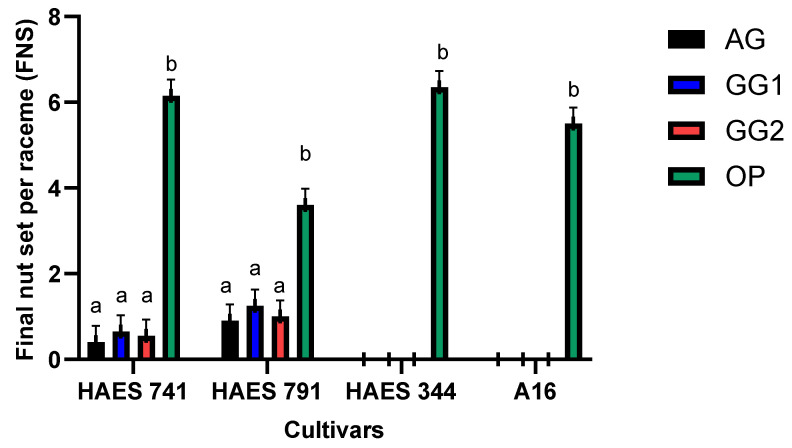
Final nut set (FNS) per raceme bars of all cultivars following different pollination methods: autogamy (AG), no hand-pollination, bagged; geitonogamy 1 (GG1), hand-pollination within the raceme, bagged; geitonogamy 2 (GG2), hand-pollination within the cultivar, bagged; open-pollination (OP), control. Bars indicate the mean FNS per raceme (±1 standard error) for the respective pollination methods. Bars marked with different letters are statistically significant (*p* < 0.05).

**Figure 3 plants-13-03456-f003:**
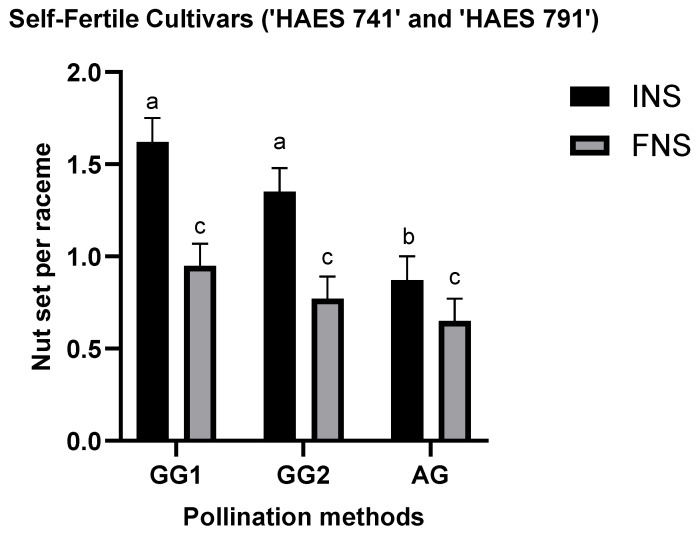
Mean initial nut set (INS) and final nut set (FNS) per raceme bars for self-fertile macadamia cultivars (‘HAES 741’ and ‘HAS 791’) following different self-pollination methods: autogamy (AG), no hand-pollination, bagged; geitonogamy 1 (GG1), hand-pollination within the raceme, bagged; and geitonogamy 2 (GG2), hand-pollination within the cultivar, bagged. Bars indicate the mean INS and FNS per raceme (±1 standard error) for the respective self-pollination methods. Bars marked with different letters (within each nut set stage) are statistically significant (*p* < 0.05).

**Figure 4 plants-13-03456-f004:**
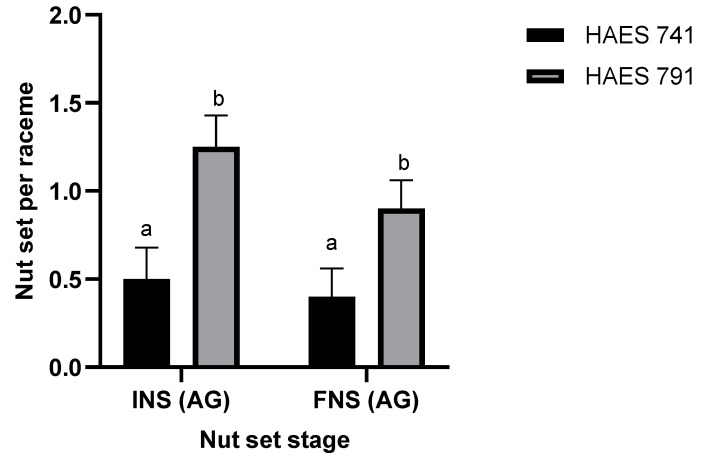
Mean initial nut set (INS) and final nut set (FNS) per raceme for self-fertile macadamia cultivars (‘HAES 741’ and ‘HAS 791’) following autogamy (AG), no hand-pollination. Bars indicate the mean INS and FNS per raceme (±1 standard error). Bars marked with different letters are statistically significant (*p* < 0.05).

**Figure 5 plants-13-03456-f005:**
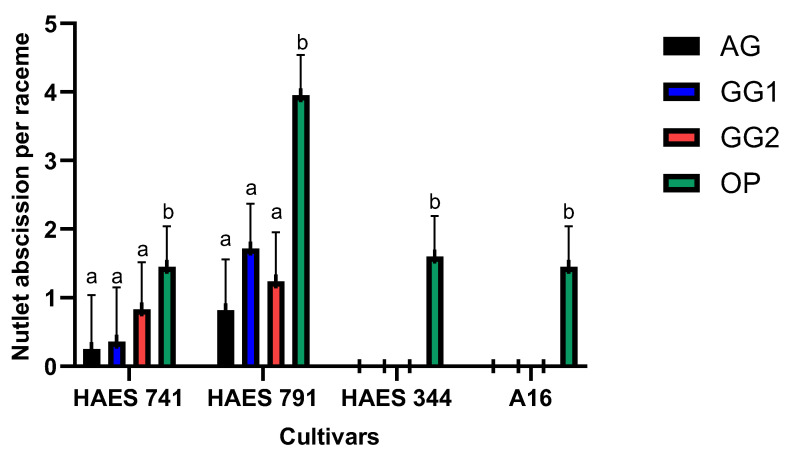
Nutlet abscission per raceme of the all cultivars following different pollination methods: autogamy (AG), no hand-pollination, bagged; geitonogamy 1 (GG1), hand-pollination within the raceme, bagged; geitonogamy 2 (GG2), hand-pollination within the cultivar, bagged; and open-pollination (OP), control. Bars indicate the mean nutlet abscission per raceme (±1 standard error) for the respective pollination methods. Bars marked with different letters are statistically significant (*p* < 0.05).

**Figure 6 plants-13-03456-f006:**
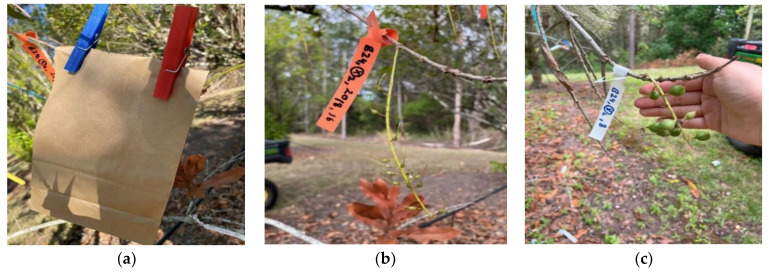
Raceme of cultivar ‘HAES 791’ at different experimental stages. (**a**) Bagging after anthesis; (**b**) initial nut set (INS); (**c**) final nut set (FNS).

**Table 1 plants-13-03456-t001:** Wald test statistics for initial nut set (INS) and final nut set (FNS) of the three self-pollination methods (AG, GG1, and GG2) among self-fertile cultivars (‘HAES 741’ and ‘HAES 791’).

Traits	Source of Variation	Wald Statistic	df	*p*-Value
INS	Cultivar	42.86	1	<0.001
	Method	16.45	2	0.019
	Cultivar × Method	9.21	2	0.061
FNS	Cultivar	14.34	1	0.009
	Method	3.25	2	0.273
	Cultivar × Method	0.21	2	0.902

**Table 2 plants-13-03456-t002:** Wald statistics for nutlet abscissions of different self-pollination methods (AG, GG1, and GG2) among self-fertile cultivars (‘HAES 741’ and HAES 791’).

Source of Variation	Wald Statistic	df	*p*-Value
Cultivar	5024	1	0.062
Method	3.95	2	0.222
Cultivar × Method	1.73	2	0.482

**Table 3 plants-13-03456-t003:** Fertility status and replication of the four study cultivars.

Fertility	Cultivars	Replication	References
Self-Fertile	HAES 741	2	[1]
	HAES 791	2	[16]
Self-Infertile	HAES 344	2	[1]
	A16	2	[16]

**Table 4 plants-13-03456-t004:** Monthly weather data for the period from September 2022 to January 2023 covering the phases from the start of pollination to the final nut set count in the macadamia trial located at Maroochy Research Facility, Nambour.

Month	T.max (°C)	T.min (°C)	Rain (mm)	Humidity (%)
September	26.2	10.8	104.2	74.7
October	33.8	10.5	168.5	77.8
November	34.2	8.4	58.0	66.4
December	35.7	15.3	76.3	74.7
January	34.7	15.6	126.1	76.3

Abbreviations: T.max, maximum temperature; T.min, minimum temperature; rain, total rainfall; relative humidity: humidity.

## Data Availability

Data will be available upon request. Please contact m.alam@uq.edu.au.

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
