# Peer review of "Evaluating Self-Pollination Methods: Their Impact on Nut Set and Nutlet Abscission in Macadamia"

_plants, 2024, doi:10.3390/plants13243456_

Round 1
Reviewer 1 Report
Comments and Suggestions for Authors
1. The manuscript is loaded with data aberrations that limit the fluence of reading and understanding.
2. The main issue of the manuscript is nuts abscissions after self-pollination. The data of two self-incompatibility cultivars should end after the initial fruit set figure.
3. the initial fruit set figure should be added before the final fruit set. the abscission rate is between INS and FNS!
4. the manuscript fails to explain why the authors think GG1 should differ from GG2 (or AG) in fruit abscission. the explanation given (row 144) is related to pollination success, rather than abscission.
5. the first mention of GG1 and GG2 (row 79) should include an explantion of the pollination method.
Reviewer 2 Report
Comments and Suggestions for Authors
"Evaluating Self-Pollination Methods: Their Impact on Nut Set and Nutlet Abscission in Macadamia", This manuscript addresses the critical topic of self-pollination in macadamia, focusing on the nut set and nutlet abscission across different self-pollination methods. The study is well-conceived and provides valuable insights into the reproductive biology of macadamia, especially the potential benefits and limitations of self-pollination for orchard productivity. However, there are several areas that need clarification or improvement to strengthen the paper's impact.
1.The introduction provides a comprehensive background but could be streamlined. Some details on the reproductive biology of macadamia could be condensed to maintain focus on the research objectives. Clearly state the research hypothesis at the end of the introduction to guide readers.
2. Why selected this four cultivars? It’s might not be sufficient to represent the entire macadamia germplasm. A broader range of cultivars could provide more comprehensive insights into the self-pollination behavior of macadamia.
3. The random selection of 10 racemes per tree may introduce sampling bias. Consider a more systematic sampling method to ensure better representation of the tree's pollination characteristics.
4. The statistical analysis could benefit from additional post-hoc tests to further explore the differences between the self-pollination methods and their interactions with cultivars.
5. The implications of the results for macadamia breeding programs could be discussed in more detail. Specifically, how the self-fertility trait can be effectively incorporated into breeding strategies to improve overall orchard productivity.
6.The description of self-pollination methods is thorough. However, it might be helpful to clarify how bagging minimizes contamination and whether any potential limitations were considered. For reproducibility, include more details on environmental conditions during the experiment (e.g., temperature, humidity, and the presence of potential insect vectors).
7.The discussion is well-written but could be enhanced by comparing your results with those of similar studies in other self-incompatible crops. The study could have benefited from a comparison with previous research on macadamia pollination to better contextualize the findings and identify areas of agreement or disagreement.
8. While the study covers relevant aspects of macadamia reproduction, it would be beneficial to more explicitly emphasize how your findings advance existing knowledge, especially in the context of breeding programs and orchard management. Consider highlighting how these results can guide future breeding efforts or influence pollination management practices.
9. The statistical analyses are robust, but the interpretation of the results, particularly concerning nut set differences between self- and open-pollination methods, could be expanded. Discuss the biological implications of these differences in more depth. For example, the discussion could benefit from more exploration of why 'HAES 741' and 'HAES 791' exhibit partial self-fertility and how this trait might be leveraged in practical scenarios.
10. The figures are informative but could be made more reader-friendly. Consider adding annotations or brief explanatory notes directly on the figures to clarify key observations. In Figure 1, adding a comparison or a reference line that emphasizes the significant differences between pollination methods could make the trends more apparent.
11. In the Figure 1/2/3, should the significant difference be calculated and marked with asterisks or a, b, c?
Round 2
Reviewer 1 Report
Comments and Suggestions for Authors
the changes made by the authors are satisfying.